# Value and Unmet Needs in Non-Invasive Human Papillomavirus (HPV) Testing for Oropharyngeal Cancer

**DOI:** 10.3390/cancers13030562

**Published:** 2021-02-02

**Authors:** Alec J. Kacew, Glenn J. Hanna

**Affiliations:** 1Biological Sciences Division, Pritzker School of Medicine, The University of Chicago, 924 E 57th St, Chicago, IL 60637, USA; akacew@uchicago.edu; 2Dana-Farber Cancer Institute, 450 Brookline Ave, Boston, MA 02215, USA

**Keywords:** human papillomavirus, head and neck cancer, oropharyngeal cancer, vaccination, biomarkers, personalized medicine, virology, oncology, public health, molecular diagnostics, immunology

## Abstract

**Simple Summary:**

As the leading human papillomavirus (HPV)-related cancer, oropharyngeal cancer places a significant burden on patients, families, and health systems. Techniques to easily and quickly test people for HPV through non-invasive means (saliva or blood tests) could, in principle, help us better understand this disease, prevent it, and treat it. However, there is currently no standardized methodology for testing saliva or blood for HPV, and such testing is not a part of routine clinical practice. In this review, we discuss and compare some of the collection and testing platforms that researchers have studied to date. We also evaluate the potential strengths and limitations of these technologies for addressing unmet needs in specific clinical contexts.

**Abstract:**

The burden of human papillomavirus (HPV)-related oropharyngeal cancer (OPC) has risen, now representing the most common HPV-related malignancy. For years, researchers have explored the utility of measuring HPV-related markers from mouth, throat, and blood samples, often with the aim of gathering more information about an existing HPV-related tumor in a given patient. We review the widely varying methods for collecting and testing saliva and blood samples and offer guidance for standardizing these practices. We then review an array of clinical contexts in which non-invasive testing holds the most promise for potentially addressing unmet needs. In particular, such testing could help clinicians and researchers monitor the effects of vaccination and treatment. Meanwhile, due to the currently incomplete understanding of how carrying HPV relates to infection and subsequent oncogenesis, non-invasive testing methods may not be suitable for the screening setting at this time.

## 1. Introduction

Human papillomavirus (HPV)-associated oropharyngeal cancer (OPC), with an estimated 12,600 new cases per year in the United States, now outpaces all other HPV-associated malignancies by incidence (HPV-associated cervical cancer is next most common, with an estimated 9700 new cases per year) [1]. For decades, while the rates of HPV-negative oral and oropharyngeal cancers have declined, the rates of HPV-positive disease have been on the rise [2]. While the disease is most frequently diagnosed in whites and in males, mortality rates are similar across demographic groups [3]. Across all stages of disease and smoking status, individuals with HPV-associated OPC have a hazard ratio 1.5–2.5 of death at 10 years following diagnosis compared to age-matched peers [4]. As with most cancers, prognosis declines with increasing stage of disease, implying that earlier diagnosis improves outcomes.

Unlike many other cancers, OPC is associated with a distinct, well-characterized, relatively homogeneous etiology in most cases (>70%): HPV infection [1]. This direct connection between microbe and pathogenesis offers a particular opportunity to harness biomarkers (including the early (E6, E7) and late (L1) oncogenic proteins uniquely expressed by HPV) for the purpose of improving our understanding, prevention, diagnosis, and treatment of the disease [5]. Although the advent of vaccination has represented a leap forward in achieving some of these goals, the burden of disease is still significant, and many affected individuals do not attain a so-called “cure.” Like other markers that are used in the context of head and neck cancers (e.g., Epstein-Barr virus (EBV), epidermal growth factor receptor (EGFR), HPV-related markers are advantageous in that they can be detected via non-invasive methodologies (namely, mouth, throat, and blood sampling). In the course of such testing, high-risk or oncogenic strains of HPV (most commonly 16, 18, 31, 33, and 35) must be differentiated from benign strains (such as 1–4, 6, and 11).

In this review, we compare the methodologies that researchers have most commonly examined for sample collection and analysis of HPV among selected studies in recent literature. Among the expanse of research available on the topic of non-invasive HPV testing, we chose to focus our discussion on later studies (2017 and onward). While we chose, when possible, to discuss studies with large sample sizes, we also consider smaller studies that offer exceptional insights (e.g., rare head-to-head data). We limit our scope to the most commonly studied sample collection and testing methodologies. We discuss the value that HPV testing may offer in a range of specific public health and clinical scenarios related to HPV-associated OPC.

## 2. HPV Detection Methods

### 2.1. Mouth and Throat Samples

#### 2.1.1. Collection Methods

Oral rinse is a common method for collecting samples from the mouth and throat (Table 1). Other approaches have included brushings from tonsillectomy specimens, oral swabs, oropharyngeal brushings (with or without endoscopic assistance), and oropharyngeal swabs. In head-to-head studies comparing collection methods, oral rinse tends to detect HPV at higher or similar rates compared to alternative techniques [6,7,8,9,10]. Researchers in one study observed higher rates of HPV detection in spit samples using a commercial collection kit compared to oral rinse samples, although these results varied by specific genetic testing approach [11]. In this study, all participants performed the oral rinse first and the spit test within one minute after the rinse. This methodology leaves open the possibility that more HPV genetic material is present following an oral rinse compared to no preceding rinse.

Among oral rinse sample studies, there is substantial variation in specific approach to this collection method. Generally, participants held the collection fluid (10–15 mL Listerine, Scope, or saline) in their mouths for 15–30 s [6,7,8,9,10,11,12,13,14,15,16,17,18]. This time was usually split into periods of gargling and periods of rinsing, with exact timing varying by study. Study reports offer a range of details about specific oral rinse procedures, with some reports providing no details beyond the fact that oral rinse samples were obtained. No studies to date have compared different oral rinse methodologies with one another.

What is unclear, given these varying methodologies, is whether some methodologies detect more HPV due to greater sensitivity or due to lower specificity. That mystery leads us to a dilemma: what constitutes clinically meaningful HPV infection? Does the mere presence of HPV-related material in the mouth or throat mean that an individual has a potentially pathogenic infection? The fact that several studies demonstrated higher HPV rates in oral rinse samples compared to brushing and swabbing samples may indeed indicate that presence of HPV in oral rinse is not the same as infection of cells that may be collected with brushes or swabs. This question warrants further research.

Future studies involving oral HPV biomarkers should provide ample detail relating to sample collection methodologies to allow for greater standardization and comparability across studies. Investigation involving head-to-head comparisons of varying oral rinse approaches is an unmet need and may help to provide insight on the question of viral presence versus persistent infection.

#### 2.1.2. Testing Methods

Investigators performed HPV detection and, in some cases, quantification, using polymerase chain reaction (PCR)-based molecular analysis. Primers were designed to identify one or more of the HPV E6, E7, and L1 oncogenes. Investigators commonly used the Roche Linear Array HPV genotyping platform for non-quantitative HPV detection. Some studies incorporated reverse transcriptase (RT)-PCR, quantitative PCR (qPCR), or a combination of the two (RT-qPCR), while others made use of the more recently developed droplet digital PCR (ddPCR) method (Figure 1). Although none of our selected clinically oriented studies directly compared ddPCR to other methodologies, current scholarship suggests that ddPCR is a more sensitive approach than traditional methodologies, as it may detect lower copy numbers of genetic material [24]. Here, again, we face the question of whether ultra-high “sensitivity” is a desired characteristic of a non-invasive HPV test. What is true “infection” and do miniscule amounts of HPV genetic material offer a reliable indication of infection (or viral integration) that might promote neoplasia? Uncovering answers to such questions is vital for elucidating the clinical utility of non-invasive HPV testing.

Although less common than molecular testing of mouth and throat samples, there has been some exploration of HPV-directed antibodies in these samples [14]. We address possible applications of such methodologies in our discussion of clinical scenarios.

### 2.2. Serum Samples

We observe two general approaches to serum testing: molecular analysis of circulating tumor DNA (ctDNA) and enzyme-linked immunosorbent assay (ELISA)-based analysis of HPV-targeted antibodies. In a study that compared ELISA-based detection of E6 antibodies with oral HPV16 DNA detection, nominal sensitivity of the antibody test was substantially higher (88% versus 43–51%), with similar specificity (≥98%) [16]. However, the fact that concordance of serum and saliva results were extraordinarily low (only one participant, 0.3%, had HPV16 detected by both methods) gives us reason for concern. Perhaps mucosal infection does not reliably generate a systemic immunologic response, suggesting that serum testing may not be the most dependable assessment tool in some settings. It should be noted that the discordance between HPV in saliva and in serum samples is not uniform across studies. Among studies that collected matched oral and serum samples and tested them for anti-HPV antibodies, some discovered a correlation between oral and serum samples and others found no correlation [25,26,27]. Among studies that assessed matched oral and serum HPV DNA, one found that each type of sample offered distinct prognostic and predictive information, while another concluded that combining results from oral and serum testing offered the greatest predictive value [17,28]. Further work is needed to understand which information is more valuable in which situations and how results from oral/oropharyngeal samples relate to those from serum samples.

## 3. Utility of Detection: Public Health and Clinical Scenarios

### 3.1. Non-Invasive Testing to Study Epidemiology of Oral/Oropharyngeal HPV

Sensibly, studies aimed at understanding the global epidemiology of HPV infection in the absence of a cancer diagnosis make use of mouth and throat testing rather than the slightly more invasive blood testing. Estimates of the prevalence of HPV genetic material in the mouth and throat in relatively unselected populations across geographies range from 1% to over 15% [11,12,15,29,30,31,32,33]. Estimates in higher risk populations (including HIV-infected men and individuals undergoing tonsillectomy for non-oncologic reasons) offer an even greater range: less than 1% to nearly 50% [6,7,8,13]. To be sure, this variability may be attributed in some part to true differences in prevalence in different regions. However, the extent to which sample collection and testing methodologies play a role is unclear. Non-invasive diagnostic tools may be useful in the future for international studies that can use a standardized methodology to better understand the extent of, and explanation for, geographical disparities in the presence of mouth and throat HPV.

Interestingly, one study demonstrated that the concordance of HPV presence and strain between bodily sites (mouth versus penis versus anus) was low [13]. This finding suggests that epidemiological efforts must consider HPV status of various anatomical sites separately. Indeed, even testing neighboring sites (e.g., oral cavity and oropharynx) can yield substantial discordance and the possibility of co-infection with various HPV strains is plausible [8].

Broad epidemiological studies using non-invasive HPV biomarker testing could reveal yet-unrealized risk factors for HPV-related disease. For example, we observe that the prevalence of mouth and throat HPV seems to be higher in individuals undergoing tonsillectomy compared to other settings [6]. Roughly two-thirds of tonsillectomies in this study were related to an infectious indication (most commonly respiratory viruses and group A streptococci). Might this discovery suggest that tonsil or other upper airway infections promote infection with HPV? Are these HPV infections persistent or transient? Large-scale non-invasive HPV biomarker epidemiological studies, if they include thorough surveys and longitudinal data collection, could provide answers to questions like these.

### 3.2. Immunogenicity

Parker and colleagues demonstrated that both mouth/throat and serum HPV-directed antibodies can offer information that correlates to some degree with timing since HPV vaccination [14]. In their study, both oral and serum anti-HPV16 and 18 antibodies rose and declined in the 30 months following initiation of HPV vaccination series. This knowledge points towards a rich domain for future research. Namely, following oral and serum antibody levels over longer periods of time in a larger group of study subjects, while simultaneously screening these individuals for HPV-related cancers, could help us uncover information about the potential utility of booster shots, for example, or could help us understand which groups of individuals might demonstrate higher risk due to lower immunogenicity (which could impact vaccination schedules). These tests could also be useful as parts of clinical trials with HPV vaccines that may be developed in the future, given that discovering modified dosing schedules is already an active area of research [34]. A greater understanding of the role for vaccination in immunocompromised groups such as survivors of prior cancers and individuals with human immunodeficiency virus is also an unmet need that immunogenicity investigation might address [35,36]. As post-marketing studies provide more information about the long-term effectiveness of the vaccines, we will gain a greater understanding of the importance and/or utility of monitoring immunogenicity over time [37].

### 3.3. Screening

Key issues that must be addressed to assess the utility of screening healthy individuals for the presence of HPV in saliva or serum include how results would impact clinical decision-making and whether screening could be cost-effective.

Although, as expected, carrying oral HPV is associated with a greater risk for OPC (nearly 50 times the odds for a person carrying HPV16 versus someone who does not) [10], the incidence of OPC (37 per 10,000 each year) is far lower than the prevalence of oral HPV (3.5%) [15]. Some older studies found even less of a difference in risk for oral or oropharyngeal cancers among individuals with evidence of oral HPV versus those without HPV (one study found that prevalence of oral HPV was similar among individuals with oral or oropharyngeal cancers and healthy individuals) [29,30,38]. The more recent figures imply that, of those who tested positive for oral HPV, roughly one in ten would go on to develop OPC. Cost-effectiveness analyses would need to evaluate the utility and cost of testing and of various monitoring techniques (for example, periodic imaging or nasopharyngoscopic exams) following a positive oral HPV test result. Clinical studies would need to determine the degree to which such diagnostic interventions impacted outcomes. Such analyses would also need to consider whom to screen and how often.

Older non-invasive biomarker studies helped to establish the basis for our current understanding of risk factors for HPV-related oropharyngeal cancers. Knowledge gained from this scholarship could help determine which groups might be the most appropriate candidates for screening programs. Included on the lengthy list of (now well-accepted) risk factors determined by these studies to increase the risk of OPC are male sex, increased number of lifetime vaginal sex partners, and increased number of lifetime oral sex partners [30,39]. These studies also helped to delineate HPV-related OPC as an etiologically separate entity from non-HPV-related OPC, as tobacco and alcohol use are associated with non-HPV-related disease but not with HPV-related disease. Oral rinse, oral brush, and serology testing for anti-E6, E7, and/or L1 antibodies were among the most common methods used by earlier studies.

Interestingly, the results of one more recent study, consistent with prior work, indicate that carrying oral HPV does not correlate with abnormal cytologic findings [9,38]. This discovery sheds some doubt on the utility of HPV testing as a screening tool to identify individuals at higher risk for developing OPC. Mitigating considerations include the possibility that the study of 310 subjects was underpowered to detect connections between virus and cytologic abnormalities and that non-invasive collection may not yield representative oropharyngeal cells that are suitable for cytologic evaluation [40].

Although the approach of using serum antibody testing as a screening tool is less common than testing orally collected specimens in the literature, there is some evidence that this strategy could be associated with higher accuracy compared with oral testing [16]. However, the best sensitivity observed in this study was 88% for HPV16 E6 antibodies. This value may not reach the standard for which one might hope from a prospective screening test.

In all, non-invasive HPV detection would have a high bar to meet in terms of cost-effectiveness, given that a minority of patients with evidence of HPV infection will go on to develop OPC and that there are some open questions about the utility and accuracy of testing methods in this setting. Before (and if) non-invasive HPV testing could be useful in screening, a greater understanding of the implications of carrying HPV (versus a clinically meaningful HPV infection) would be needed.

### 3.4. Monitoring Treatment Effect

While quantifying HPV infection through non-invasive testing at OPC diagnosis does not seem to offer much prognostic value, there are promising signals of benefit associated with non-invasive HPV testing as part of monitoring treatment effect [41]. The commercially available ddPCR circulating tumor DNA (ctDNA) NavDx^®^ (Naveris, Waltham, MA, USA) platform has demonstrated clinically meaningful utility in tracking response or outcomes to chemoradiotherapy [23]. Individuals with clinical risk factors and poor clearance of HPV16 ctDNA after definitive treatment had a 35% actuarial rate of treatment failure, a significant difference from the ctDNA clearance group, of whom, none of them experienced treatment failure (recognizing the number of failure events was small in the study). Our group has seen similar results with our validated ddPCR assay, with which we were able to predict imaging findings based on trends in HPV cell-free DNA [17,42]. These results can help clinicians tailor treatment. For example, a rise in HPV cell-free DNA during definitive therapy could prompt an escalation or change in treatment regimen. Inversely, an early clearance in HPV could flag a patient as a candidate for treatment de-escalation. Such dynamic, real-time therapy alterations could represent an innovation in personalized care of individuals with HPV-related OPC.

### 3.5. Surveillance for Recurrence

Our prior work has demonstrated that the natural course of salivary HPV-targeting antibodies is to decline over time following completion of definitive therapy for OPC [43]. Beyond on-treatment monitoring, the NavDx^®^ technology has also shown potential in the setting of surveillance following completion of definitive therapy. The test demonstrated a negative predictive value of 100% in anticipating disease-free status among individuals with undetectable HPV ctDNA at all post-treatment timepoints and demonstrated a 94% positive predictive value for recurrence among individuals with two consecutive positive results after completing therapy [22]. The second positive result predated biopsy-proven recurrence by a median of nearly four months. Given the chance for sustained survival in a minority of individuals undergoing salvage surgery for recurrent oropharyngeal disease, earlier detection of recurrence, and thus earlier salvage surgery, could have a significant impact [44]. Similar results to the ones in the NavDx^®^ trial have been observed in other studies using different PCR-based technologies and other collection methods (including non-invasively collected sample from tumor brushings) [18,19,21]. Still, another study has provided an example of how HPV ctDNA data can be combined with radiologic data to develop even more accurate predictions about recurrence [20]. A number of these investigations were included in a meta-analysis that calculated an area under the receiver-operator curve of 1.0 for the pooled value of cell-free DNA in predicting recurrence [45].

## 4. Conclusions

Non-invasive HPV testing has the potential to be of great use in certain public health and clinical contexts related to HPV-associated OPC. However, greater understanding of the pathogenic mechanisms linking HPV exposure to HPV infection and those linking HPV infection to OPC tumorigenesis are prerequisites to fully harnessing the value of non-invasive testing. For HPV testing to be useful in the screening setting, clinicians would require additional information besides the presence or absence of HPV genetic material or of immune response to HPV. Detecting potentially oncogenic infection using markers of gene expression (e.g., messenger ribonucleic acids corresponding to the HPV E6, E7, or L1 genes or validated host cell markers like p16) is a ripe area for study in the screening setting. Once we can appreciate which markers are most meaningful for predicting oncogenesis, we will be in a position to conduct head-to-head trials to understand which tissue collection and laboratory testing methodologies provide the most accurate measures of those markers. After comparative studies have provided indications of optimal collection and testing techniques, clinician researchers will then be able to work out how best to use those diagnostic tools to guide clinical decisions in the screening setting. In sum, there is currently insufficient data to support the use of non-invasive HPV testing in the screening setting. Clinical data demonstrating improved outcomes resulting from non-invasive screening would be a requirement for this strategy to prove useful. At present, the clinical implications of positive screening test results are unclear.

Testing immunogenicity following vaccination can help optimize vaccine development and public health strategies aimed at vaccination. We recommend that future vaccine clinical trials incorporate immunogenicity testing to understand if some groups of patients could benefit from more or fewer doses. Trialists could also attempt longer follow-up (e.g., on the order of decades) to gain a greater understanding of how markers of immunogenicity correlate with observed risk of HPV infection and HPV-associated malignancy.

Monitoring HPV levels during definitive therapy could, in the future, help personalize treatment. We could hypothesize, for example, that individuals whose HPV markers drop precipitously early on in definitive therapy might be candidates for de-escalation (e.g., in the form of a shorter course of radiotherapy). Inversely, those with persistently elevated HPV markers might benefit from a more extended course of definitive therapy or novel adjuvant regimens. Incorporating non-invasive HPV testing as part of routine monitoring in clinical trials studying definitive therapy can help us uncover such fresh approaches. Incorporating non-invasive HPV testing into follow-up after definitive therapy in these trials could offer insight into how such testing could help detect recurrences earlier. Even only with currently available data, we believe that there is a role for plasma HPV ctDNA platforms as an adjunct to traditional response metrics (i.e., imaging) in the context of monitoring response and surveilling for recurrence. Although research to date has focused on treatment with curative intent, future studies should examine the utility of monitoring HPV in the recurrent/metastatic setting. Especially as our understanding of how HPV causes OPC improves, non-invasive HPV testing promises to offer innovations in the field of HPV-related OPC.

## Figures and Tables

**Figure 1 cancers-13-00562-f001:**
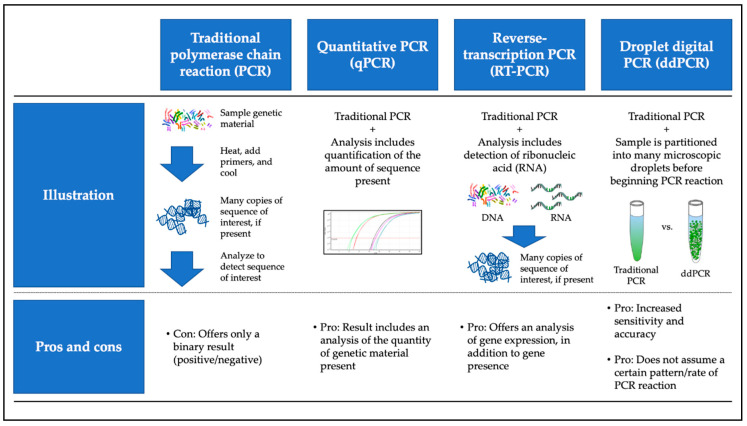
Overview of most commonly used molecular techniques for non-invasive HPV biomarker testing. Includes color images available from Wikimedia as part of the creative commons license.

**Table 1 cancers-13-00562-t001:** Selected recent studies of non-invasive HPV testing.

First Author	Year	N	Setting	Collection Method	Testing Method	Genotypes Included	Findings
**Mouth/Throat Sample Collection**							
Stankiewicz Karita [12]	2020	15,313	Epidemiology, Seattle, Washington, United States	Oral rinse	RT-qPCR	16, 18	1% overall prevalence, OR 3.2 for men vs. women
Kofler [19]	2020	62	Surveillance for recurrence	Endoscopic oropharynx brushing	RT-PCR	Broad (40 types)	Clearance of oropharyngeal HPV DNA predicts lower chance of recurrence
Benevolo [9]	2020	310	Screening for HPV-related malignancy	Oral rinse vs. brushing	PCR	Broad (37 types)	HPV genetic material does not correlate with cytologic abnormalities
Dona [8]	2019	163	Epidemiology, Rome, Italy, high-risk individuals	Oral rinse vs. oropharyngeal brushing vs. oral brushing	PCR	Broad (37 types)	51.2% agreement for oral rinse vs. oropharyngeal brushings, 74.1% for high-risk genotypes
D’Souza [16]	2019	694	Screening for HPV-related malignancy	Oral rinse, serum	PCR, ELISA	Broad	Low sensitivity (43–88%) and high specificity (≥98%) in the screening setting; serum Ab testing performs better than oral rinse
Hanna [17]	2019	21	Risk stratification after OPC diagnosis, monitoring treatment	Oral rinse, serum	ddPCR	16, 18, 31, 33, 45	Baseline plasma ctHPVDNA levels associated with poor outcomes; trends in salivary DNA predicts outcomes
Fakhry [18]	2019	396	Surveillance for recurrence	Oral rinse	PCR	Broad (37 types)	Detection of oral HPV after therapy portends worse RFS and OS
Chikandiwa [7]	2018	181	Epidemiology, Johannesburg, South Africa, HIV-infected men	Paired oral rinse vs. oral swab	PCR	Broad (37 types)	1.8% prevalence in oral rinse vs. 0.6% in oral swab
Tsikis [13]	2018	294	Epidemiology, Athens, Greece, high-risk men	Oral rinse vs. anal swab vs. penile swab	Next-generation sequencing	Broad	49% prevalence at any site: 33% anal, 23% penile, 4% oral; Low concordance (≤2%) between oral and anogenital site
De Souza [11]	2018	96	Epidemiology, Brisbane, Australia	Oral rinse vs. spit (commercial saliva kit)	PCR (single primer vs. nested)	Broad	Oral rinse: 11.5% (nested PCR), 10.4% (single primer PCR) Spit: 16.7% (nested PCR), 3.1% (single primer PCR)
Combes [6]	2017	692	Epidemiology, France	Oral rinse vs. brushing from tonsillectomy specimen	PCR (bead-based multiplex assay)	Broad (21 types)	13.1% prevalence in rinse vs. 3.6% in tonsil brushings
D’Souza [15]	2017	13,089	Epidemiology, screening for HPV-related malignancy	Oral rinse	PCR	Broad (37 types)	3.5% prevalence of HPV infection, 37 per 10,000 annual OPC incidence
Laprise [10]	2017	918	Screening for HPV-related malignancy	Oral rinse vs. brushing	PCR	Broad (37 types)	HPV infection associated with OR 10.8 for OPC, 47.2 with HPV16 infection
**Serum-Based Sample Collection**							
Tanaka [20]	2020	35	Surveillance for recurrence	Serum (ctDNA)	ddPCR	16	ctHPV16DNA, when combined with PET-CT, predicts recurrence
Reder [21]	2020	50	Surveillance for recurrence	Serum (ctDNA)	RT-qPCR	16	Lower post-therapy ctHPVDNA corresponds with reduced chance of recurrence
Chera [22]	2020	115	Surveillance for recurrence	Serum (ctDNA)	ddPCR	16, 18, 31, 33, 35	Undetectable ctHPVDNA at all post-treatment timepoints has 100% NPV for recurrence; two consecutive positive ctHPVDNA tests after treatment has 94% PPV for recurrence
Chera [23]	2019	103	Monitoring treatment	Serum (ctDNA)	ddPCR	16, 18, 31, 33, 35	Poor ctHPVDNA clearance associated with treatment failure; ctHPVDNA copy number associated with tumor burden and HPV genome integration
**Includes both Mouth/Throat and Serum-Based Samples**							
Parker [14]	2019	150	Immunogenicity, international, adult males receiving quadrivalent vaccine	Matched oral rinse, serum at multiple timepoints	ELISA	16, 18	Oral anti-HPV Abs present in majority at month 7, minority at month 18

Ab = antibody, ctDNA = circulating tumor deoxyribonucleic acid, ddPCR = droplet digital polymerase chain reaction, ELISA = enzyme-linked immunosorbent assay, HPV = human papillomavirus, NPV = negative predictive value, OPC = oropharyngeal cancer, OR = odds ratio, OS = overall survival, PCR = polymerase chain reaction, PPV = positive predictive value, qPCR = quantitative polymerase chain reaction, RFS = recurrence-free survival, RT = reverse transcription, WGS = whole genome sequencing.

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
