# Peer review of "Value and Unmet Needs in Non-Invasive Human Papillomavirus (HPV) Testing for Oropharyngeal Cancer"

_cancers, 2021, doi:10.3390/cancers13030562_

Round 1

Reviewer 1 Report

The authors present a review about non-invasive HPV-testing for oral cancer. A series of current studies (n=18) is listed in table 1, and the references reflect the current literature.

The authors draw the conclusion that non-invasive HPV-testing offers the largest benefit in treatment monitoring and testing for recurrence. 

In their abstract, the authors stated that they 'offer guidance for standardizing these practices'. In my opinion there is a bit of a shortfall in guidance, as the authors give no recommendation for standardizing the HPV-testing.

Is the colored figure taken from Wikipedia? This should be mentioned in the corresponding legend.

Author Response

We thank the reviewer for these thoughtful comments. Please see below for point-by-point response and attached for tracked changes version of the revised manuscript. Our responses are marked in bold and line numbers refer to those in the “tracked changes” version.

The authors present a review about non-invasive HPV-testing for oral cancer. A series of current studies (n=18) is listed in table 1, and the references reflect the current literature.

The authors draw the conclusion that non-invasive HPV-testing offers the largest benefit in treatment monitoring and testing for recurrence.

In their abstract, the authors stated that they 'offer guidance for standardizing these practices'. In my opinion there is a bit of a shortfall in guidance, as the authors give no recommendation for standardizing the HPV-testing.

We have expanded the conclusions section to clarify our point of view about how investigators can, in the future, harness non-invasive testing in the screening setting (lines 280-296). We propose a three-step process:

  1. Through laboratory research, identify which markers predict progression from HPV exposure to HPV infection and then to eventual oncogenesis
  2. Conduct head-to-head studies of collection and testing methods, with the primary outcome measure being accuracy in measuring the marker(s) identified in step 1
  3. Conduct clinical trials to elucidate the optimal application of the collection/testing method identified in step 2 to clinical decision-making

We have also expanded the discussion of how investigators can study non-invasive testing in other clinical settings (lines 304-328).

Is the colored figure taken from Wikipedia? This should be mentioned in the corresponding legend.

We have credited Wikimedia in the figure legend (lines 130-131) and have kept the original credit written in the “acknowledgements” section.

Reviewer 2 Report

The authors gives an excellent review about this issue

I recommend only to give more clears conclussions, for example what do you recommends in the clinical practice to do about HPV determinations?

Author Response

We thank the reviewer for the thoughtful comment. Please see below for point-by-point response and attached for tracked changes version of the revised manuscript. Our responses are marked in bold and line numbers refer to those in the “tracked changes” version.

The authors gives an excellent review about this issue

I recommend only to give more clears conclussions, for example what do you recommends in the clinical practice to do about HPV determinations?

In our view, given the current level of knowledge in the field, there is not a role for non-invasive HPV testing in today’s clinical practice. Rather, we outline what additional information would be needed for non-invasive testing to be relevant for clinical practice.

We have expanded the conclusions section to clarify our point of view about how investigators can, in the future, harness non-invasive testing in the screening setting (lines 280-296). We propose a three-step process:

  1. Through laboratory research, identify which markers predict progression from HPV exposure to HPV infection and then to eventual oncogenesis
  2. Conduct head-to-head studies of collection and testing methods, with the primary outcome measure being accuracy in measuring the marker(s) identified in step 1
  3. Conduct clinical trials to elucidate the optimal application of the collection/testing method identified in step 2 to clinical decision-making

We have also expanded the discussion of how investigators can study non-invasive testing in other clinical settings (lines 304-328).

Reviewer 3 Report

The Authors present a review of HPV screening studies in OPC, including the specificity, limitations and scope of various methodologies, from sample collection to analysis. They highlight potential for improvement and the need for consistency in future studies, and the problem of detection with respect to clinical relevance. This is all of great interest in moving forward with public health screening programs, ironing out the problems with biomarker selection and logistics.

The review is well written, interesting and covers the scope described in the title.

Author Response

We are grateful for the reviewer’s kind and encouraging feedback. Please see attached for a "tracked changes" version of the revised manuscript.

Reviewer 4 Report

The summary and abstract are really succinct and support the body of the paper.

The introduction is strong and brings into view the issues that current researchers and clinicians are dealing with in the field.

Table 1 presents "selected" studies and one wonders how these selections were made.  I would like to know why these studies were selected and why so many other studies were not selected.  Some studies have miniscule "N".  Recent studies were obviously preferred.  What about studies before 2017?

Nevertheless, this data is important and it is presented to the reader in a format that is easy to assimilate.

The section introducing oral sample collection methodology is very well done and I was surprised that sampling methods have not been compared .  

Testing methods and Fig.1:  Very nice and clear.

Serum samples: Too much emphasis is put on the study where there was no concordance between serum and saliva testing, since the authors state that this is not necessarily a representative result and other studies exist that contradict this result.  Pull up those other studies, give us a summary.

Utility of testing section is excellent and rich in information.  

Conclusion section:  The first sentence: Change the word "to" to "the" to read "Non-invasive testing has the potential...".

The remaining part of this section is excellent and succinct.

Author Response

We thank the reviewer for these thoughtful comments. Please see below for point-by-point response and attached for tracked changes version of the revised manuscript. Our responses are marked in bold and line numbers refer to those in the “tracked changes” version.

The summary and abstract are really succinct and support the body of the paper.

The introduction is strong and brings into view the issues that current researchers and clinicians are dealing with in the field.

Table 1 presents "selected" studies and one wonders how these selections were made.  I would like to know why these studies were selected and why so many other studies were not selected.  Some studies have miniscule "N".  Recent studies were obviously preferred.  What about studies before 2017?

We now offer, at the end (lines 58-63) of the introduction, some explanation for how we chose our highlighted studies in Table 1.

Nevertheless, this data is important and it is presented to the reader in a format that is easy to assimilate.

The section introducing oral sample collection methodology is very well done and I was surprised that sampling methods have not been compared . 

Testing methods and Fig.1:  Very nice and clear.

Serum samples: Too much emphasis is put on the study where there was no concordance between serum and saliva testing, since the authors state that this is not necessarily a representative result and other studies exist that contradict this result.  Pull up those other studies, give us a summary.

In lines 142-147, we now offer an overview of other studies that assess matched oral and serum samples.

Utility of testing section is excellent and rich in information. 

Conclusion section:  The first sentence: Change the word "to" to "the" to read "Non-invasive testing has the potential...".

We are grateful to the reviewer for having caught this typo. We have corrected it (line 280).

The remaining part of this section is excellent and succinct.

Round 2

Reviewer 1 Report

The manuscript is much better after the revision.